# Communication-efficient distributed eigenspace estimation with arbitrary node failures

**Vasileios Charisopoulos**
Operations Research & Information Engineering
Cornell University
Ithaca, NY 14853
vc333@cornell.edu

**Anil Damle**
Computer Science
Cornell University
Ithaca, NY 14853
damle@cornell.edu

## Abstract

We develop an eigenspace estimation algorithm for distributed environments with arbitrary node failures, where a subset of computing nodes can return structurally valid but otherwise arbitrarily chosen responses. Notably, this setting encompasses several important scenarios that arise in distributed computing and data-collection environments such as silent/soft errors, outliers or corrupted data at certain nodes, and adversarial responses. Our estimator builds upon and matches the performance of a recently proposed non-robust estimator up to an additive $\tilde{O}(\sigma\sqrt{\alpha})$ error, where $\sigma^2$ is the variance of the existing estimator and $\alpha$ is the fraction of corrupted nodes.

## 1 Problem overview and background

Modern machine learning has seen the proliferation of heterogeneous distributed environments for training and deploying data science pipelines. As communication between machines is often the most time-consuming operation in distributed systems, the design of *communication-efficient algorithms* is of paramount importance for scaling to massive datasets [36]. However, the move to distributed environments also adds several additional layers of complexity in the design of algorithms. For example, in the distributed setting we would like our algorithms to be *robust* and providing meaningful answers even in settings where some nodes contain outlier data [4], silently fail during the computation [27, 31], or are compromised and returning malicious results designed to corrupt the central solution.

This work focuses on distributed *eigenspace estimation* in the context of robustness to node-level corruptions. Formally, we assume a computing environment with nodes numbered $i = 1, \ldots, m$, where every node $i$ observes a local version $A_i$ of an unknown symmetric matrix $A \in \mathbb{R}^{d \times d}$; the goal is to approximate the subspace spanned by the $r \ll d$ principal eigenvectors of $A$. **Distributed PCA** is a standard example in this framework: every machine draws $n$ i.i.d. samples from an unknown distribution $\mathcal{P}$ with covariance matrix $A$ and forms a local empirical covariance matrix $A_i$. Recently proposed communication-efficient algorithms have every node $i$ transmit $V_i$, the $d \times r$ matrix of principal eigenvectors of $A_i$, to a central server, which then aggregates all the local solutions via a carefully-crafted aggregation procedure [10, 18].

We devise and analyze an algorithm that is robust to a wide range of corruptions that can occur to a subset of the computational nodes. In particular, we assume that some fraction $\alpha$ of the computational nodes can respond with completely arbitrary, but structurally valid, responses (i.e., they return arbitrary matrices $V_i$ with orthonormal columns). This model encompasses three common forms of node-level corruption that cannot be easily detected by the central machine in isolation:

**Silent/soft errors.** While computational errors may be rare on single machines, as distributed workloads span large numbers of nodes the probability that some of them fail becomes significant.

Though catastrophic failures may be detectable, allowing the central server to simply ignore the output of specific nodes, the more nefarious issue is that of so-called silent (or soft) errors [16, 19, 27]. More specifically, a silent error is one where a node returns an erroneous but structurally valid response to the central machine query. Because the response is structurally valid and the central machine may not have access to the per-node data it is not possible to "validate" the response of each node and, instead, the central estimator must be adapted to be robust to such errors.

**Outliers or corrupted data.** In certain settings the data collection may be distributed in addition to the computation. If some of the nodes are drawing samples from an invalid or corrupted data source they may introduce gross outliers to the set of responses $\{V_i \mid i \in [m]\}$. Similarly, in the distributed PCA example, while most machines draw a sufficient number of samples, a minority of them may have only a small amount of data available such that the principal eigenspaces of the local empirical covariance matrices are too far from the ground truth, and thus violate standard modelling assumptions in distributed learning. Again, robustness to such outlier responses must be a feature of the estimator since they cannot be detected by individual nodes (as they do not have information about the global problem).

**Adversarial responses.** In some settings, a subset of nodes may be compromised by an adversary who wishes to influence the central solution by crafting and returning malicious $V_i$. In fact, the adversarial nodes may be collaborating when constructing their responses. Since the central node does not get to see all the data it cannot validate responses or directly detect adversaries. Therefore, the estimator itself must be adapted to be robust to collections of responses designed to push the solution in specific directions.

The main contribution of our paper is a *communication-efficient algorithm that is robust to node corruptions* (as outlined above) for the distributed eigenspace estimation problem. We note that our corruption model is similar to so-called Byzantine failures [25] in distributed systems.

## 1.1 Related work

**Distributed eigenspace estimation.** The problem of distributed eigenspace estimation has been well-studied in the absence of malicious noise. One of the challenges in the distributed setting is aggregating local solutions in the presence of symmetry: for example, if $v$ is an eigenvector of $A$, both $\pm v$ are valid solutions to our problem. Various works deal with such symmetries in different ways; in the algorithms of [5, 18], the central node averages the *spectral projectors* of the local eigenspaces, and performs an eigendecomposition of the resulting average to approximate the principal eigenspace. This approach is similar to the algorithms of [3, 11, 26], although the latter works focus on distributed *low-rank approximations* and do not address the issue of approximating the principal eigenspace directly. Another standard approach is for the central server to aggregate local solutions after an alignment step designed to remove the orthogonal ambiguity [10, 17, 21] (see also [7] for the non-distributed setting). Indeed, our work builds on the two-stage algorithm presented and analyzed in [10] for the non-robust setting. Finally, we briefly mention a recent line of work [12, 21] that adapts the *shift-and-invert preconditioning* framework [20] to the distributed setting; however, the latter approach leads to algorithms that require multiple rounds of communication.

**Robust PCA.** The literature contains a number of different formulations for robust principal component analysis. The seminal work of Candés et al. [9] formulated robust PCA as the task of separating an observed matrix $Y \in \mathbb{R}^{d \times d}$ into a low-rank and a sparse component – a slightly different problem from that considered in this paper. Xu et al. [34] considered the problem of approximating a low-dimensional distribution from a set of $n$ i.i.d. samples, a constant fraction of which have been individually corrupted by gross outliers. Follow-up works focused on high-dimensional sparse estimation and applications to sparse robust PCA [2, 15]. However, communication-efficient robust PCA appears to be overlooked. A notable exception is the sketch-based algorithm of [30], although the assumptions therein depart significantly from our setting. Another relevant line of work is that of Byzantine-robust distributed learning (typically focusing on distributed gradient descent); see, e.g., [1, 6, 8, 13, 14, 24, 29, 35] as well as the survey [23]. In these works, an iterative algorithm is distributed across machines that send individual updates to a central server, which combines them using a robust aggregation procedure (e.g., the geometric median [29]). Such works are more general in scope, but typically lead to estimators that require multiple rounds of communication. For minimization of strongly convex quadratics, [35] proposes a one-shot algorithm employing a robust

aggregation scheme. However, this special case departs from our setting as their underlying problem is convex and solved under the assumption that solutions are unique.

## 1.2 Notation

We let $\mathbb{S}^{d-1}$ denote the unit sphere in $d$ dimensions. We write $\|A\|_{\mathrm{F}} := \sqrt{\langle A, A \rangle}$ and $\|A\|_2 := \sup_{x \in \mathbb{S}^{d-1}} \|Ax\|_2$ for the Frobenius and spectral norms of a matrix $A \in \mathbb{R}^{n \times d}$. We write $\mathbb{O}_{n,r}$ for the set of $n \times r$ matrices with orthonormal columns and $\mathbb{O}_r \equiv \mathbb{O}_{r,r}$. Given $U, V \in \mathbb{O}_{n,r}$ we write

$$\mathrm{dist}(U, V) := \left\| (I - UU^\mathsf{T})V \right\|_2 = \left\| (I - VV^\mathsf{T})U \right\|_2 \tag{1}$$

for their $\ell_2 \to \ell_2$ subspace distance and $\mathcal{B}_{\mathrm{dist}}(U; r)$ for the scaled unit ball centered at $U$:

$$\mathcal{B}(U; r) := \{V \in \mathbb{O}_{d,r} \mid \mathrm{dist}(U, V) \le r\}.$$

Finally, we use the notation $A \lesssim B$ to indicate that $A \le cB$ for a dimension-independent constant $c > 0$ and $A \asymp B$ if $A \lesssim B$ and $B \lesssim A$ simultaneously.

## 2 Robust distributed eigenspace estimation

We now formally introduce the problem setting. In particular, we assume there exists an *unknown* symmetric matrix $A$ with spectral decomposition

$$A = V \Lambda V^\mathsf{T} + V_\perp \Lambda_\perp V_\perp^\mathsf{T}, \quad V \in \mathbb{O}_{n,r}, \ \Lambda = \mathrm{diag}(\{\lambda_i(A)\}_{i=1}^r), \ \Lambda_\perp = \mathrm{diag}(\{\lambda_i(A)\}_{i=r+1}^d), \tag{2}$$

assuming a nonincreasing ordering on the eigenvalues:

$$\lambda_1(A) \ge \cdots \ge \lambda_r(A) > \lambda_{r+1}(A) \ge \cdots \ge \lambda_d(A).$$

Our goal is to approximate the principal $r$-dimensional eigenspace $\mathcal{V} := \mathrm{span}(V)$ of $A$ given $m$ machines, each of which observes a local version $A_i$ of $A$, communicating with a central coordinator. We assume that $m$ is even for simplicity. When queried for a response, machine $i$ responds either with an eigenvector matrix spanning the principal eigenspace of the local matrix $A_i$, or with an arbitrary $d \times r$ matrix with orthonormal columns. The latter case corresponds to so-called *compromised* machines. In contrast, prior work [10, 18] assumes that every machine responds truthfully.

**Assumption 1** (Corruption model). There exists a constant $\alpha \in (0, 1/2)$ and an index set $\mathcal{I}_{\mathsf{bad}} \subset [m]$ with $|\mathcal{I}_{\mathsf{bad}}| / m \le \alpha$ such that the following holds: all nodes $i \notin \mathcal{I}_{\mathsf{bad}}$ observe a symmetric matrix $A_i \in \mathbb{R}^{d \times d}$. Moreover, when queried for a response, every node $i$ returns

$$\widehat{V}_i = \begin{cases} V_i, & i \in [m] \setminus \mathcal{I}_{\mathsf{bad}}, \\ Q_i, & i \in \mathcal{I}_{\mathsf{bad}}, \end{cases} \tag{3}$$

where the columns of $V_i \in \mathbb{O}_{d,r}$ span the principal $r$-dimensional eigenspace of $A_i$ and $Q_i \in \mathbb{O}_{d,r}$ is an arbitrary $d \times r$ matrix with orthonormal columns.

For notational convenience, we also define the set of "good" responses:

$$\mathcal{I}_{\mathsf{good}} := [m] \setminus \mathcal{I}_{\mathsf{bad}}, \quad \text{with} \quad |\mathcal{I}_{\mathsf{good}}| \ge (1 - \alpha)m. \tag{4}$$

Furthermore, we require the principal eigenspace of $A$ to be sufficiently separated from its complement and that the local errors $E_i := A_i - A$ are not too large.

**Assumption 2.** There is a constant $\delta > 0$ such that the following hold:

1. **(Gap)** The matrix $A$ has a nontrivial eigengap:

$$\delta_r(A) := \lambda_r(A) - \lambda_{r+1}(A) \ge \delta. \tag{5}$$

2. **(Approximation)** For all $i \in \mathcal{I}_{\mathsf{good}}$, the local observations satisfy:

$$\|A_i - A\|_2 \le \frac{\delta_r(A)}{8}. \tag{6}$$

The problem admits a natural proxy of difficulty in the form of the *normalized inverse eigengap* $\kappa$:

$$(\textbf{Normalized inverse eigengap}) \qquad \kappa := \frac{\|A\|_2}{\delta_r(A)} = \frac{\lambda_1(A)}{\lambda_r(A) - \lambda_{r+1}(A)}. \qquad (7)$$

Our algorithm for the robust distributed eigenspace estimation problem is outlined in Algorithm 1, which is essentially a robust version of the Procrustes fixing algorithm from [10]. The latter (non-robust) algorithm operates as follows: first, every machine $i$ computes its local eigenvector matrix $V_i$ and broadcasts it to the central server. Because invariant subspaces do not admit unique representations, naively averaging these estimates can fail to reduce the approximation error further. Instead, the algorithm of [10] first picks one of the local solutions (say $V_1$) as a reference and "aligns" every other solution with it by solving a so-called *Procrustes problem*:

$$Z_i := \operatorname*{argmin}_{U \in \mathbb{O}_r} \|V_i U - V_1\|_{\mathrm{F}}, \quad i = 2, \ldots, m. \qquad (8)$$

After the alignment step (8), the solution of which is available in closed form via the SVD [22], the central coordinator computes and returns the empirical average $(1/m)\sum_{i=1}^m V_i Z_i$.

## 2.1 Technical challenges and main result

To robustify the algorithm described above against node failures, we need the following ingredients:

**Reference estimation.** In the presence of corruptions one must guard against the possibility of choosing an outlier as a reference solution (which would render the alignment step (8) useless). The first step of our algorithm robustly determines a reference guaranteed to have nontrivial alignment with the ground truth (Algorithm 2).

**Solution aggregation.** With the robust reference at hand, the next step of the algorithm aligns other local solutions with it. However, since some of the solutions are outliers, we use a robust mean estimation algorithm in the last step of Algorithm 1 to compute the empirical average only over inliers (and possible "benign" outliers) with high probability.

Combining the above ingredients requires working around additional technical challenges. The first challenge is showing that averaging over inliers that have been aligned with $\widehat{V}_{\mathsf{ref}}$ is equivalent (up to small error) to averaging over inliers that have been aligned with the ground truth $V$. To do so, we leverage a path-independence result from [33] to view the aligned estimates as the eigenvectors arising from a carefully chosen sequence of perturbations to the unknown matrix $A$ (Lemma 1).

After aligning local estimates, we approximate the empirical mean over inliers with a spectral filtering algorithm. Existing analyses of that algorithm focus on vector-valued inputs and lead to error bounds in the Euclidean norm (or other vector norms, see [32]). Invoking these error bounds in a black-box fashion for matrix-valued inputs (by flattening every eigenvector matrix to a $(dr)$-dimensional vector) has multiple drawbacks. On one hand, it leads to unnatural scaling as the final result depends on the empirical covariance over flattened inputs. On the other hand, the resulting error bounds are with respect to the Frobenius norm, while the spectral norm is a standard error measure in eigenspace estimation. To avoid this, we modify the iterative filtering algorithm so that inputs are not flattened and obtain error bounds with respect to the spectral norm (3).

We analyze each ingredient of Algorithm 1 separately, in Sections 2.2 to 2.4; all proofs appear in the supplementary material. Our analysis is almost completely deterministic: indeed, the only source of randomness is the filtering algorithm used in the final stage (Algorithm 5).

---

**Algorithm 1** Robust distributed eigenspace estimation

---

**Input**: responses $\left\{\widehat{V}_i\right\}_{i=1,\ldots,m}$, corruption fraction $\alpha$, failure prob. $p$, error parameter $\omega$.

$\widehat{V}_{\mathsf{ref}} := \texttt{RobustReferenceEstimator}\left(\widehat{V}_1, \ldots, \widehat{V}_m\right)$.  $\triangleright$ Algorithm 2; Section 2.2

$\left\{\widetilde{V}_i\right\}_{i=1,\ldots,m} := \texttt{ProcrustesFixing}\left(\left\{\widehat{V}_1, \ldots, \widehat{V}_m\right\}, \widehat{V}_{\mathsf{ref}}\right)$  $\triangleright$ Algorithm 3; Section 2.3

$\bar{V} := \texttt{AdaptiveFilter}\left(\left\{\widetilde{V}_1, \ldots, \widetilde{V}_m\right\}, 6, \omega, p, \alpha\right)$  $\triangleright$ Algorithm 5; Section 2.4

**return** $\bar{V}$

---

Our main theorem on the performance of Algorithm 1 follows below. The result is first stated for inputs from a generic eigenspace estimation problem. In Section 3, we apply it in a black-box fashion to specialize its results for distributed PCA.

**Theorem 1.** *Let Assumptions 1 and 2 hold and suppose that the corruption level $\alpha$ satisfies*

$$\varphi := \alpha + \frac{6\log(1/p)}{m} < \frac{1}{12}. \tag{9}$$

*For a fixed failure probability $p$ and error parameter $\omega$, Algorithm 1 returns $\bar{V} \in \mathbb{R}^{d \times r}$ satisfying:*

$$\operatorname{dist}(\bar{V}, V) \lesssim \underbrace{\frac{1}{\delta}\left\|\frac{1}{|\mathcal{I}_{\text{good}}|}\sum_{i \in \mathcal{I}_{\text{good}}} A_i - A\right\|_2}_{E_{\text{oracle}}} + \underbrace{\frac{\kappa^2}{|\mathcal{I}_{\text{good}}|}\sum_{i \in \mathcal{I}_{\text{good}}}\left(\frac{\|A_i - A\|_2}{\delta}\right)^2}_{E_{\text{high}}} + \underbrace{\sqrt{\varphi \max(\omega, \sigma^2)}}_{E_{\text{robust}}}. \tag{10}$$

*with probability at least $1 - 2\log(6/\omega) \cdot p$. Moreover, the variance $\sigma^2$ satisfies*

$$\sigma^2 \leq \left\|\frac{1}{|\mathcal{I}_{\text{good}}|}\sum_{i \in \mathcal{I}_{\text{good}}} V_i V_i^\mathsf{T} - V V^\mathsf{T}\right\|_2 + 2 \cdot \left\|\frac{1}{|\mathcal{I}_{\text{good}}|}\sum_{i \in \mathcal{I}_{\text{good}}} \widetilde{V}_i - V\right\|_2. \tag{11}$$

**Remark 1** (Error bound interpretation). *The error bound (10) decomposes naturally into three terms that can be controlled individually in applications. The first term, $E_{\text{oracle}}$, corresponds to an oracle estimator that approximates $V$ via the principal eigenspace of $1/|\mathcal{I}_{\text{good}}|\sum_{i \in \mathcal{I}_{\text{good}}} A_i$. The second term, $E_{\text{high}}$, represents high-order errors that occur as a result of the alignment step in Algorithm 3 that are dominated by $E_{\text{oracle}}$ in "typical" situations. Finally, the term $E_{\text{robust}}$ quantifies the effect of outliers, vanishing as the fraction of corrupted nodes $\alpha \downarrow 0$. We comment on the scaling of $E_{\text{robust}}$ relative to the error of the non-robust algorithm in the context of distributed PCA in Section 3.*

## 2.2 The robust reference estimator

This section focuses on the analysis of Algorithm 2, which yields the robust reference estimator $\widehat{V}_{\text{ref}}$ used to remove the orthogonal ambiguity from local solutions. We note that the construction of the estimator dates back to the seminal work of Nemirovski and Yudin [28].

---

**Algorithm 2** RobustReferenceEstimator$(Y_1, \ldots, Y_m)$

---

    **for** $i = 1, \ldots, m$ **do**
        $\varepsilon_i := \min\left\{r \geq 0 \mid |\mathcal{B}_{\text{dist}}(Y_i; r) \cap \{Y_i\}_{i=1}^m| > \frac{m}{2}\right\}$
    **return** $Y_{i_\star}$, where $i_\star := \operatorname{argmin}_{i \in [m]} \varepsilon_i$

---

**Remark 2.** *The quantities $\varepsilon_i$ in Algorithm 2 can be found in time $O(m^2 dr^2)$ by first computing $r_j := \operatorname{dist}(Y_i, Y_j)$ for all $j \neq i$ and setting $\varepsilon_i := \mathtt{median}(\{r_j\}_{j \neq i})$.*

Note that even though $\widehat{V}_{\text{ref}}$ could be chosen among some of the compromised samples, its construction ensures that it inherits the accuracy of the majority of the responses.

**Proposition 1** (Robust reference estimator). *Given a sample $\{Y_1, \ldots, Y_m\}$ where $Y_i \in \mathbb{O}_{d,r}$ and $|\{i \in [m] \mid \operatorname{dist}(Y_i, V) \leq \varepsilon\}| > m/2$ for a fixed $\varepsilon > 0$, Algorithm 2 outputs $Y_{i_\star}$ satisfying*

$$\operatorname{dist}(Y_{i_\star}, V) \leq 3\varepsilon. \tag{12}$$

## 2.3 The ProcrustesFixing algorithm

In this section, we formally introduce the Procrustes-fixing procedure and show that it properly aligns all the non-compromised responses given the reference solution described in Section 2.2. The procedure is described in Algorithm 3; it accepts a set of $d \times r$ matrices with orthonormal columns as well as a reference matrix $Y_{\text{ref}}$ of the same shape.

The work [10] provides an error bound for the ProcrustesFixing algorithm under idealized conditions; namely, that the reference solution is equal to the ground truth $V$.

---

**Algorithm 3** ProcrustesFixing($\{Y_1, \ldots, Y_m\}, Y_{\mathsf{ref}}$)

---

    **for** $i = 1, \ldots, m$ **do**
      $\widetilde{Y}_i := Y_i Z_i, \quad$ where $\quad Z_i := \operatorname{argmin}_{Z \in \mathbb{O}_r} \|Y_i Z - Y_{\mathsf{ref}}\|_{\mathrm{F}} \qquad\qquad$ ▷ Procrustes alignment
    **return** $\left\{\widetilde{Y}_i \mid i \in [m]\right\}$

---

**Theorem 2** (Theorem 2 in [10]). *Let Assumption 2 hold and let*

$$\tilde{V} := \frac{1}{|S|} \sum_{i \in S} \widetilde{V}_i, \quad \text{where} \quad \left\{\widetilde{V}_i\right\}_{i \in S} = \texttt{ProcrustesFixing}(\{V_i\}_{i \in S}, V), \ S \subset \mathcal{I}_{\mathsf{good}}.$$

*Then the following bound holds:*

$$\left\|\tilde{V} - V\right\|_2 \lesssim \frac{1}{\delta^2} \frac{1}{|S|} \sum_{i \in S} \|A_i - A\|_2^2 + \frac{1}{\delta} \left\|\frac{1}{|S|} \sum_{i \in S} A_i - A\right\|_2. \tag{13}$$

While the setting of Theorem 2 is idealized, when the reference chosen by Algorithm 2 is sufficiently close to $V$ one would expect that the aligned estimates are not far from their ideal version. The next Lemma shows that aligning the local solutions with $\widehat{V}_{\mathsf{ref}}$ is equivalent to aligning with the ground truth $V$, up to higher-order errors.

**Lemma 1.** *Let $V_i \in \mathbb{O}_{d,r}$ span the principal $r$-dimensional eigenspace of the matrix $A_i$ and let $V \in \mathbb{O}_{d,r}$ span the principal $r$-dimensional invariant subspace of $A$. Suppose that there is a $V_{\mathsf{ref}} \in \mathbb{O}_{d,r}$ satisfying $\operatorname{dist}(V_{\mathsf{ref}}, V) = \varepsilon < \delta_r(A)/8$, and define the sets of aligned estimates*

$$V_i^{\mathsf{ideal}} := V_i \cdot \operatorname*{argmin}_{Z \in \mathbb{O}_r} \|V_i Z - V\|_{\mathrm{F}}, \quad V_i^{\mathsf{corr}} := V_i \cdot \operatorname*{argmin}_{Z \in \mathbb{O}_r} \|V_i Z - V_{\mathsf{ref}}\|_{\mathrm{F}}.$$

*Then for any $i \in \mathcal{I}_{\mathsf{good}}$ the following holds:*

$$\left\|V_i^{\mathsf{ideal}} - V_i^{\mathsf{corr}}\right\|_2 \lesssim \frac{1}{\delta^2} \max\left\{\|A_i - A\|_2^2, \|A\|_2^2 \, \varepsilon^2\right\}. \tag{14}$$

Putting everything together, we arrive at a *deterministic* characterization of the error attained by the empirical average over any subset of responses that come from non-compromised nodes and have been aligned with the robust reference estimator. Note that this characterization does not immediately translate to an algorithm, since the set of compromised nodes is not known a-priori.

**Proposition 2** (Error of clean samples). *Let $\widehat{V}_{\mathsf{ref}}$ be the output of Algorithm 2 given inputs $\widehat{V}_1, \ldots, \widehat{V}_m$. For any index set $S \subset \mathcal{I}_{\mathsf{good}}$ and $i \in S$, define*

$$V_i^{\mathsf{corr}} := V_i \cdot \operatorname*{argmin}_{Z \in \mathbb{O}_r} \left\|V_i Z - \widehat{V}_{\mathsf{ref}}\right\|_{\mathrm{F}}; \quad V_i^{\mathsf{ideal}} := V_i \cdot \operatorname*{argmin}_{Z \in \mathbb{O}_r} \|V_i Z - V\|_{\mathrm{F}}.$$

*Suppose that $\operatorname{dist}(V, \widehat{V}_{\mathsf{ref}}) = \varepsilon < \delta_r(A)/8$. Then the following bound holds:*

$$\left\|\frac{1}{|S|} \sum_{i \in S} V_i^{\mathsf{corr}} - V\right\|_2 \lesssim \frac{1}{\delta^2 |S|} \sum_{i \in S} \max\left(\|A_i - A\|_2^2, \|A\|_2^2 \varepsilon^2\right) + \frac{1}{\delta}\left\|\frac{1}{|S|} \sum_{i \in S} A_i - A\right\|_2. \tag{15}$$

*Proof.* From the triangle inequality, Lemma 1 and Theorem 2 it follows that

$$
\begin{aligned}
\left\|\frac{1}{|S|} \sum_{i \in S} V_i^{\mathsf{corr}} - V\right\|_2 &= \left\|\frac{1}{|S|} \sum_{i \in S} V_i^{\mathsf{corr}} - V_i^{\mathsf{ideal}} + V_i^{\mathsf{ideal}} - V\right\|_2 \\
&\leq \left\|\frac{1}{|S|} \sum_{i \in S} V_i^{\mathsf{corr}} - V_i^{\mathsf{ideal}}\right\|_2 + \left\|\frac{1}{|S|} \sum_{i \in S} V_i^{\mathsf{ideal}} - V\right\|_2 \\
&\lesssim \frac{1}{\delta^2 |S|} \sum_{i \in S} \max\left(\|A_i - A\|_2^2, \|A\|_2^2 \varepsilon^2\right) + \frac{1}{\delta}\left\|\frac{1}{|S|} \sum_{i \in S} A_i - A\right\|_2.
\end{aligned}
$$

$\qquad\qquad\qquad\qquad\qquad\qquad\qquad\qquad\qquad\qquad\qquad\qquad\qquad\qquad\qquad\qquad\qquad\qquad\square$

## 2.4 Analysis of robust mean estimation

We now analyze the last phase of the algorithm, which approximates $V$ using the robust mean of the aligned samples. The mean estimation procedure used is the randomized iterative filtering method shown in Algorithm 4, which is an appropriate modification of the spectral filtering algorithm from the robust statistics literature for matrix-valued inputs. Its analysis is available in the Appendix, while Theorem 3 below summarizes its guarantees.

---

**Algorithm 4** $\texttt{Filter}(S := \{X_i\}_{i=1,\ldots,m}, \lambda_{\mathsf{ub}})$

---

Compute empirical mean and covariance:

$$\theta_S := \frac{1}{|S|} \sum_{i \in S} X_i, \quad \Sigma_S := \frac{1}{|S|} \sum_{i \in S} (X_i - \theta_S)(X_i - \theta_S)^\mathsf{T}.$$

Compute leading eigenpair $(\lambda, v)$ of $\Sigma_S$.
**if** $\lambda < 18\lambda_{\mathsf{ub}}$ **then**
    **return** $\theta_S$
**else**
    Compute outlier scores $\tau_i := v^\mathsf{T}(X_i - \theta_S)(X_i - \theta_S)^\mathsf{T} v$ for $i \in S$.
    Sample $Z$ from $S$ following $\mathbb{P}(Z = X_i) = \frac{\tau_i}{\sum_{j \in S} \tau_j}$.
    **return** $\texttt{Filter}(S \setminus \{Z\}, \lambda_{\mathsf{ub}})$

---

**Theorem 3.** *Suppose $G_0 \subset [m]$ and that the corruption level $\alpha$ and failure probability $p$ satisfy*

$$\alpha + \frac{6\log(1/p)}{m} \leq \frac{1}{12} \quad and \quad |G_0| \geq (1-\alpha)m. \tag{16}$$

*Moreover, let $\Sigma_{G_0} := \frac{1}{|G_0|} \sum_{i \in G_0} (X_i - \theta_{G_0})(X_i - \theta_{G_0})^\mathsf{T}$, where $\theta_{G_0} := \frac{1}{|G_0|} \sum_{i \in G_0} X_i$, and fix $\lambda_{\mathsf{ub}} \geq \|\Sigma_{G_0}\|_2$. Then Algorithm 4 invoked with $\{X_i\}_{i=1}^m$ and $\lambda_{\mathsf{ub}}$ returns an estimate $\theta_{\lambda_{\mathsf{ub}}}$ satisfying*

$$\mathbb{P}\left(\left\|\theta_{\lambda_{\mathsf{ub}}} - \frac{1}{|G_0|} \sum_{i \in G_0} X_i\right\|_2 \geq 18\sqrt{5\lambda_{\mathsf{ub}}}\left(\alpha + \frac{4\log(1/p)}{m}\right)^{1/2}\right) \leq p. \tag{17}$$

**Remark 3.** *The upper bound $\alpha < 1/12$ appears to be a proof artifact and could likely be improved by optimizing choices of constants in the proof of Theorem 3. Numerical evidence in Section 3.1 suggests that the breakdown point of Algorithm 4 is closer to the natural limit of $1/2$.*

The error in (17) scales with the upper bound $\lambda_{\mathsf{ub}}$, which may be far from the "optimal" $\|\Sigma_{G_0}\|_2$. We describe an adaptive version of Algorithm 4 that achieves this at a logarithmic additional cost. Indeed, suppose an upper bound on $\alpha$ is available and the unknown parameter $\|\Sigma_{G_0}\|_2$ lies in an interval $[\lambda_{\mathsf{lb}}, \lambda_{\mathsf{ub}}]$. We construct a search grid $\mathcal{G}$ as follows:

$$\mathcal{G} := \left\{2^j \mid j \in \{j_{\mathsf{lo}}, j_{\mathsf{hi}}\}\right\}, \quad j_{\mathsf{lo}} := \lfloor\log_2(\lambda_{\mathsf{lb}})\rfloor, \quad j_{\mathsf{hi}} := \lceil\log_2(\lambda_{\mathsf{ub}})\rceil. \tag{18}$$

We are now in good shape to describe our estimator. To simplify notation, we define the error proxy

$$f(\lambda; p, \alpha) := 18\sqrt{5\lambda}\left(\alpha + \frac{4\log(1/p)}{m}\right)^{1/2}. \tag{19}$$

Our estimator, $\theta_{\hat{\lambda}}$, is implemented in Alg. 5 and defined as:

$$\theta_{\hat{\lambda}}, \text{ where } \hat{\lambda} := \min\left\{\lambda \in \mathcal{G} \mid \|\theta_\lambda - \theta_{\lambda'}\|_2 \leq f(\lambda; p, \alpha) + f(\lambda'; p, \alpha), \forall \lambda' \in \mathcal{G} \cap [\lambda, \infty)\right\}. \tag{20}$$

---

**Algorithm 5** $\texttt{AdaptiveFilter}(S = \{X_i\}_{i=1,\ldots,m}, \lambda_{\mathsf{ub}}, \lambda_{\mathsf{lb}}, p, \alpha)$

---

Set up search grid: $j_{\mathsf{lo}} := \lfloor\log_2 \lambda_{\mathsf{lb}}\rfloor, \ j_{\mathsf{hi}} := \lceil\log_2 \lambda_{\mathsf{ub}}\rceil.$
**for** $j = j_{\mathsf{hi}}, \ldots, j_{\mathsf{lo}}$ **do**
    $\theta_{2^j} = \texttt{Filter}(S, 2^j)$                                        ▷ Algorithm 4
    **if** $\exists k > j$ such that $\|\theta_{2^j} - \theta_{2^k}\| > f(2^j; p, \alpha) + f(2^k; p, \alpha)$ **then**      ▷ $f$ defined in (19)
        **return** $\theta_{2^{j+1}}$
**return** $\theta_{2^{j_{\mathsf{lo}}}}$

---

If $\|\Sigma_{G_0}\|_2 \in [\lambda_{\mathsf{lb}}, \lambda_{\mathsf{ub}}]$, the estimator attains the optimal error up to a constant while the success probability degrades only logarithmically, as shown by the following Proposition.

**Proposition 3.** *If $G_0$, $p$, and $\alpha$ satify (16) and $\|\Sigma_{G_0}\|_2 \leq \lambda_{\mathsf{ub}}$, the estimator $\theta_{\hat{\lambda}}$ from (20) satisfies*

$$\left\| \theta_{\hat{\lambda}} - \frac{1}{|G_0|} \sum_{i \in G_0} X_i \right\|_2 \leq 171 \sqrt{\max\{\|\Sigma_{G_0}\|_2, \lambda_{\mathsf{lb}}\}} \left( \alpha + \frac{4\log(1/p)}{m} \right)^{1/2}$$

*with probability at least $1 - 2\log_2(\lambda_{\mathsf{ub}}/\lambda_{\mathsf{lb}})\, p$.*

### 2.5 Proof sketch of main theorem

We briefly sketch the proof of Theorem 1 here. We decompose

$$\mathrm{dist}(\bar{V}, V) \lesssim \|\bar{V} - V\|_2 \leq \underbrace{\left\| \bar{V} - \frac{1}{|\mathcal{I}_{\mathsf{good}}|} \sum_{i \in \mathcal{I}_{\mathsf{good}}} \widetilde{V}_i \right\|_2}_{\Delta_1} + \underbrace{\left\| \frac{1}{|\mathcal{I}_{\mathsf{good}}|} \sum_{i \in \mathcal{I}_{\mathsf{good}}} \widetilde{V}_i - V \right\|_2}_{\Delta_2}$$

The error $\Delta_1$ can be directly controlled by applying Proposition 3 with $G_0 \equiv \mathcal{I}_{\mathsf{good}}$ combined with the fact that the spectral norm of the matrix $\Sigma_{\mathcal{I}_{\mathsf{good}}}$ admits the upper bound in (21); we refer the reader to the supplement for a complete derivation.

$$\|\Sigma_{\mathcal{I}_{\mathsf{good}}}\|_2 \leq \left\| \frac{1}{|\mathcal{I}_{\mathsf{good}}|} \sum_{i \in \mathcal{I}_{\mathsf{good}}} V_i V_i^\mathsf{T} - VV^\mathsf{T} \right\|_2 + 2 \left\| \frac{1}{|\mathcal{I}_{\mathsf{good}}|} \sum_{i \in \mathcal{I}_{\mathsf{good}}} \widetilde{V}_i - V \right\|_2 \tag{21}$$

Finally, we control the error $\Delta_2$ by invoking Proposition 2 with $S = \mathcal{I}_{\mathsf{good}}$, since

$$\widetilde{V}_i = V_i \cdot \operatorname*{argmin}_{Z \in \mathbb{O}_r} \left\| V_i Z - \widehat{V}_{\mathsf{ref}} \right\|_{\mathrm{F}}, \quad \text{for all } i \in \mathcal{I}_{\mathsf{good}}.$$

Combining the resulting upper bounds yields the error in Theorem 1. $\qquad\square$

**Remark 4.** *Both of the terms in (21) are typically small and can be directly controlled in concrete applications such as distributed PCA. Moreover, while the bound in (21) is not directly computable, it is immediate that $\|\Sigma_{\mathcal{I}_{\mathsf{good}}}\|_2 \lesssim 1$ and thus we may initialize Algorithm 5 with $\lambda_{\mathsf{ub}} = O(1)$ in the absence of a finer upper bound.*

## 3 Robust distributed PCA

In this section, we specialize the results of Section 2 to robust distributed PCA for subgaussian distributions. We first formalize the sampling model for the problem.

**Assumption 3** (Subgaussian data). *Every machine $i \in \mathcal{I}_{\mathsf{good}}$ draws $\{X_j^{(i)}\}_{j=1}^n \overset{\mathrm{iid}}{\sim} \mathcal{P}$, where $\mathcal{P}$ is a zero-mean, subgaussian distribution with covariance matrix $A := \mathbb{E}_{X \sim \mathcal{P}}[XX^\mathsf{T}]$, and forms $A_i := \frac{1}{n} \sum_{j=1}^n X_j^{(i)} (X_j^{(i)})^\mathsf{T}$.*

Our main theorem follows directly from Theorem 1 and control of $\|\Sigma_{\mathcal{I}_{\mathsf{good}}}\|_2$ under Assumption 3.

**Theorem 4.** *Let Assumptions 1 to 3 hold and suppose that $n \gtrsim \kappa^2 \cdot (r_\star + \log(mn/p))$ and $\alpha$, $m$ and $p$ satisfy (9). Then Algorithm 1 initialized with $\omega = \sqrt{1/mn}$ returns a $\bar{V}$ satisfying*

$$\mathrm{dist}(\bar{V}, V) \lesssim \sqrt{\varrho\left(\alpha + \frac{\log(1/p)}{m}\right)} + \kappa \sqrt{\frac{r(r_\star + \log(n))}{(1-\alpha)mn}} + \kappa^4 \cdot \frac{r(r_\star + \log(mn\log(mn)/p))}{n}, \tag{22}$$

*with probability at least $1 - 2/n - p$. Here, $r_\star := \mathrm{Tr}(A) / \|A\|_2$ and $\varrho$ is given by*

$$\varrho := \kappa \sqrt{\frac{r(r_\star + \log(n))}{(1-\alpha)mn}} + \max\{\kappa^2 \sqrt{r}, \kappa^4\} \cdot \frac{r_\star + \log(mn\log(mn)/p)}{n}.$$

When $\kappa \asymp 1$, high-order terms in Theorem 4 can be discarded and we arrive at the following:

**Corollary 1.** *Assume that the conditions of Theorem 4 hold and $\kappa \asymp 1$. Then:*

$$\text{dist}(\bar{V}, V) \lesssim \sqrt{\alpha + \frac{\log(\log(mn)/p)}{m}} \cdot \left(\frac{r(r_\star + \log(n))}{(1-\alpha)mn}\right)^{1/4} + \sqrt{\frac{r(r_\star + \log(n))}{(1-\alpha)mn}}$$

*with probability at least $1 - 2/n - p$.*

We briefly compare the error of Algorithm 1 to that of its non-robust counterpart from [10] when $\kappa \asymp 1$. The latter algorithm returns an estimate $\bar{V}^{\text{nonrobust}}$ satisfying $\text{dist}(\bar{V}^{\text{nonrobust}}, V) = \tilde{\mathcal{O}}\left(\sqrt{r_\star/mn}\right)$. Ignoring the $\sqrt{r}$ factors, which are likely an artifact of our proof, our algorithm also introduces an additive error of the order $\tilde{\mathcal{O}}(\sqrt{\alpha/1-\alpha} \cdot (r_\star r/mn)^{1/4})$. Note that for a constant absolute number of corruptions $\alpha \propto 1/m$, this additive factor scales as

$$\sqrt{\frac{\alpha}{1-\alpha}} \left(\frac{r_\star r}{mn}\right)^{1/4} \lesssim \left(\frac{r_\star r}{m^3 n}\right)^{1/4}.$$

If $m$ and $n$ are comparable, this is similar to the error of the non-robust algorithm up to an $(r/r_\star)^{1/4}$ factor. Therefore, the performance of Algorithm 1 degrades gracefully as a function of the corruption level under not too restrictive assumptions on the ratio $m/n$.

### 3.1 Numerical study

We provide a brief numerical illustration of the performance of Algorithm 1 on data sampled from an unknown Gaussian distribution $\mathcal{D} := \mathcal{N}(0, V\Lambda V^{\mathsf{T}} + V_\perp \Lambda_\perp V_\perp^{\mathsf{T}})$, where $[V \quad V_\perp] \in \mathbb{O}_d$ is a random $d \times d$ orthogonal matrix and $\Lambda, \Lambda_\perp$ are generated according to the following model:

$$\Lambda = I_r, \quad (\Lambda_\perp)_{jj} = (1-\delta)\eta^j, \quad j = 1, \ldots, d-r, \quad \text{where} \quad \eta = 1 - \frac{1-\delta}{r_\star - r} \in (0,1). \quad (23)$$

We simulate an adversary by replacing the first $\lfloor \alpha m \rfloor$ responses by the same $V_{\text{adv}} \in \mathbb{O}_{d,r}$, chosen to be near-orthogonal to $V$. We fix the gap $\delta = 0.25$ throughout. We compare Alg. 1 (labelled `Robust` in our plots) against two baselines: the algorithm from [10] (labelled `Naive`), which corresponds to Alg. 3 using the first response – which is always corrupted in our experiment – as the reference followed by naive averaging; and a version of Alg. 1 without the robust mean estimation step (labelled `Procrustes`). Our implementation always removes the sample with the largest outlier score in each step of Alg. 4 and uses a simplified error proxy $f(\lambda; \alpha) := \sqrt{\lambda \alpha}$ instead of (19) in Alg. 5.

Our experiment is illustrated in Figure 1. Clearly, the baseline methods break down in the presence of corruption, yielding solutions nearly orthogonal to $V$ as $\alpha$ approaches $1/2$. In contrast, the error of Alg. 1 degrades gracefully with $\alpha$. We note that our algorithm yields a nontrivial solution even when almost half of the measurements are corrupted ($\alpha = 45\%$), in line with intuition suggesting that $\alpha_\star = 1/2$ is a natural breakdown point for outlier-robust algorithms.

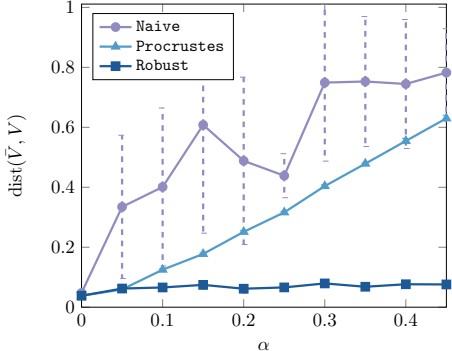
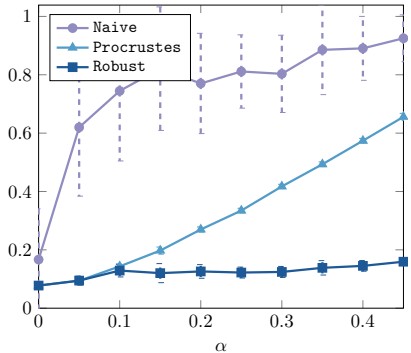

**Figure 1:** Robust distributed PCA with $m = 150$, $n = 50r$, $r_\star = 2r$, and $\kappa = 5$ under Model (23). We report the mean subspace distance $\pm$ one standard deviation over 10 independent runs for subspace dimension $r = 5$ (**left**) and $r = 10$ (**right**).

In addition, we study the effect of the number of per-machine samples $n$ and total number of machines $m$ in distributed PCA. We fix $r = 5$, $r_\star = 2r$, $\kappa = 5$ and $\delta = 0.25$ and experiment with

$$(m, n) = \left\{(32, 2^i) \mid i = 5, \ldots, 10\right\}, \quad (m, n) = \left\{(2^i, 128) \mid i = 4, \ldots, 9\right\}, \quad \alpha \in \{0.25, 0.45\}.$$

The results can be found in Fig. 2 and are consistent with our theory: for fixed $m$, a small number of samples per machine $n$ leads to inaccurate local solutions and inaccurate final estimates for all variants, while increasing $n$ leads to a decrease in error for all configurations. On the other hand, the robust algorithm produces good estimates for all attempted values of $m$ after fixing a sufficiently large sample size $n$, and increasing $m$ only decreases the error for the robust variant. Finally, we observe that the error of the non-robust variants plateaus since the corruption fraction $\alpha$ is bounded away from zero.

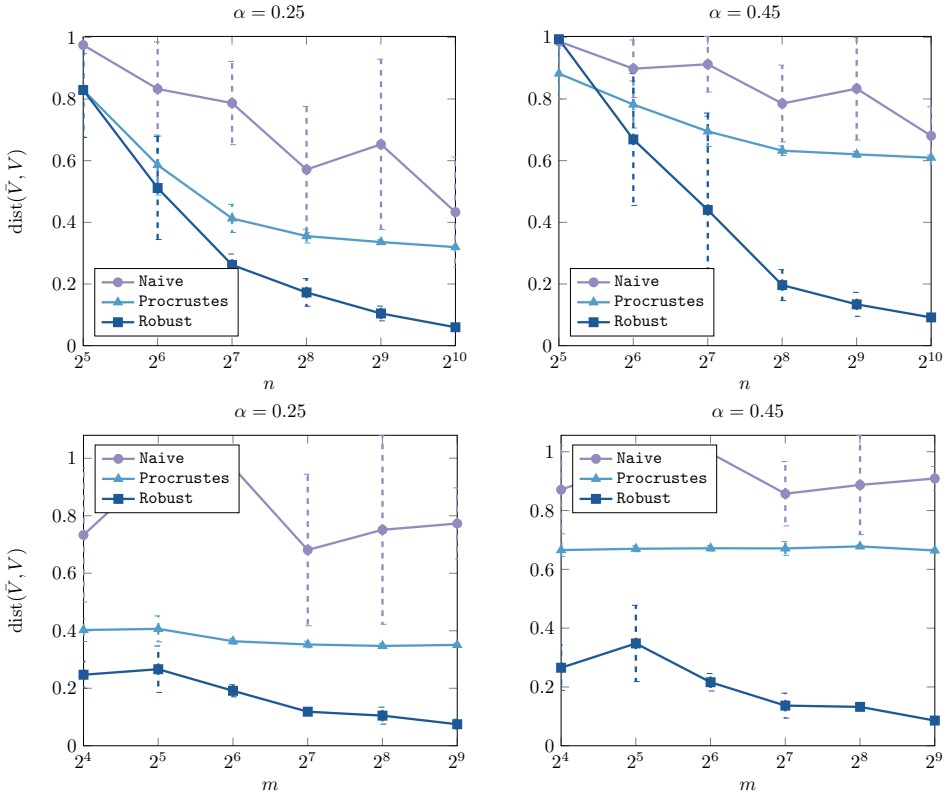

**Figure 2: Top**: Effect of sample size $n$ for fixed number of machines $m = 32$. **Bottom**: Effect of number of machines $m$ for fixed sample size $n = 128$.

## 4   Discussion

We presented a communication-efficient algorithm for distributed eigenspace estimation that is robust to $\alpha = {}^1/_{12}$ fraction of nodes returning structurally valid but otherwise potentially adversarial responses. Numerical evidence suggests its breakdown point is closer to the (optimal) $\alpha_* = {}^1/_2$, which might be achievable by an improved analysis of the filtering procedure in Alg 4. Our adaptive version of the filtering procedure in Alg 5 trades off knowledge of (an upper bound on) the corruption level $\alpha$ with the need for a tight bound on $\|\Sigma_{\mathcal{I}_{\mathrm{good}}}\|_2$. Alternatively, one can design a version of Algorithm 5 that is adaptive to the corruption level $\alpha$ using a similar construction that evaluates the error proxy $f(\lambda; p, \alpha)$ for different values of $\alpha$ and fixed $\lambda \approx \|\Sigma_{\mathcal{I}_{\mathrm{good}}}\|_2$ instead.

**Acknowledgements.**

We thank Jayadev Acharya and Damek Davis for their insightful comments and suggestions. We also thank the anonymous reviewers for invaluable feedback that greatly improved the manuscript.

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
