# OpenReview forum: "Communication-efficient distributed eigenspace estimation with arbitrary node failures"
_NeurIPS.cc/2022/Conference — NeurIPS 2022 Accept_

### Official Review · Reviewer_2GaG · 2022-07-06

**Rating:** 7
**Confidence:** 4
**Soundness:** 4 excellent
**Presentation:** 4 excellent
**Contribution:** 4 excellent

**Summary:**

The paper develops and analyzes a decentralized algorithm for eigenspace estimation in the presence of compromised nodes. This work extends the work of (Charisopoulos et al. 2021) to the robust setting, i.e., when a fraction of the nodes send faulty responses. The most interesting contribution of the paper is the performance guarantee in the form of an upper bound on the error between the true and estimated space.

**Questions:**

* It would help to clarify where the loosness of the bound is coming from. Is it from the robust mean estimation component? You could probably add some thoughts to the discussion regarding stronger techniques to tighten the bounds.
* Please make the theorems self-contained by defining the symbols in the statement of the theorem.
* More numerical results could be added to the supplement.

**Strengths And Weaknesses:**

Strengths
+ The proof is elegant and very well written. This is a solid paper.
+ Developing a robust version of the procrustes algorithm is commended given the emergence of robust learning.
+ The ideas are compelling. In particular, (i) since a fraction of the measurements is corrupted, a reliable reference is constructed to be used for alignment by choosing the index corresponding to the smallest radius of sufficiently large balls around the received sample containing more than half of the machines; (ii) robust mean estimation is adequately used to aggregate the local versions given the presence of outliers.
+ The different ingredients of the algorithm are analyzed in a nice progression leading to a deterministic analysis, which is then used to give high probability bounds for subgaussian distributions.

Weaknesses (most are minor)
- While the paper provides performance guarantees, the sufficient condition is loose. It requires the fraction of compromised nodes to be even smaller than 1/12. The numerical results show sustained performance even when half of the nodes are corrupted.
- There is non-cited work on communication efficient robust PCA in the distributed setting. This is one example:
 M. Rahmani and G. Atia, “A Decentralized Approach to Robust Subspace Recovery,” Allerton 2015.
While the approaches are different this is relevant and the claim in the Related Work section should be corrected.
- Some symbols are never defined in the body of the theorem and the reader has to look carefully at the parameters passed to the functions in the algorithms to understand what they represent (examples: p, omega, \Sigma_G0, ...).
- The numerical results section is a bit limited and can be improved for further validation.

---

> ### Author Response · Authors · 2022-08-01
> **Response to Reviewer 2GaG**
>
>
> Thank you for the positive review and thoughtful feedback!
>
> We have added the missing reference to the Related Work section, and are currently working on making all theorems self-contained and removing unnecessary notation. Due to
> the 9-page limit, some of this restructuring will have to be deferred to the camera-ready version (assuming our paper gets accepted). We apologize for the inconvenience.
>
> We address your major questions and concerns individually below.
>
> ## On the looseness of the sufficient condition
>
> The looseness of the condition on $\alpha$ is likely an artifact of the proof
> for the robust mean estimation algorithm, which introduces arithmetically
> convenient restrictions for expressions involving $\alpha$ in several steps.
> While one could trace
> every step of the proof and try to optimize for $\alpha$, this will be tedious
> and could lead to an unnecessarily long proof, especially if the goal is to
> get $\alpha$ arbitrarily close to $1/2$.
>
> In fact, we already spent a considerable amount of time trying to pin down
> an explicit value for the upper bound on $\alpha$, since the reference whose
> analysis we are adapting only stated a condition of the form
>
> $$
> \alpha + \frac{\log(1 / p)}{m} < c,
> $$
>
> albeit without specifying a value for $c$. Our main takeaway of the theory, for the time being, is that one can tolerate a *constant* fraction of
> corruptions (as opposed to a  fraction that decays in $m$).
>
> We have updated the revised manuscript to suggest that the looseness is a proof
> artifact that could be improved. However, we are not aware of a proof technique
> that would lead to a better constant without the need to optimize constants in
> multiple steps.
>
> ## More numerical experiments
>
> Thank you for the suggestion. We have added several additional figures in the
> supplementary material in response to yours and other reviewers' concerns. The new experiments examine the effect of sample size $n$, number of machines $m$, and an alternative method for selecting the reference solution borrowed from Ref. \[8\] in the manuscript.

---

### Official Review · Reviewer_GRdZ · 2022-07-10

**Rating:** 7
**Confidence:** 2
**Soundness:** 3 good
**Presentation:** 3 good
**Contribution:** 3 good

**Summary:**

This paper considers a setting where $m$ nodes collaborate via a server to (approximately) compute the eigenspace spanned by the $r$ principal eigenvectors of a matrix $A$. Each node has access to i.i.d. samples of $A$, and the goal is to exchange information among the nodes to compute an estimate of $A$ that is "better" than what each node could compute in isolation. The key challenge tackled in the problem is the aspect of *security*: a fraction $\alpha$ of the nodes is adversarial and can act arbitrarily. The problem then is to still provide meaningful results for the estimation of $A$ that scale gracefully with the corruption fraction $\alpha$. The authors address this problem by developing a robust distributed eigenspace estimation algorithm that requires only one round of communication; rigorous convergence bounds are also established for this algorithm.

**Questions:**

I have no further questions beyond the clarifying questions I posed earlier.

**Limitations:**

I do not see any negative societal impacts of this work.

**Strengths And Weaknesses:**

I found the paper quite well-written in general; the main algorithmic ideas and proof techniques are explained in a very clear way. My main comments are as follows.

*Strengths*
- Although adversarially robust distributed optimization/statistical learning has been studied in several prior works, the specific formulation considered in this paper is novel: I have not seen robust mean estimation ideas used in the context of distributed eigenspace estimation before. The analysis of the proposed algorithm is also solid, making the overall approach a meaningful contribution.

The comments I have are mainly to clarify a few issues, but nothing major. I should also point out here that I am not an expert on eigenspace estimation.

*Comments*
- Coming back to the aspect of novelty, the contribution of the paper would be further strengthened in my opinion if the authors could articulate more elaborately why their algorithmic ideas/analysis are not subsumed by the numerous prior works on Byzantine-robust distributed optimization. What exactly makes their setting different/challenging? Can't the eigenspace estimation problem also be cast as an optimization problem? What precludes this line of approach?

Right now, the authors mention in passing that the existing works on robust distributed optimization/learning typically employ iterative algorithms that require multiple rounds of communication between the master and the worker nodes. In contrast, their approach requires only one round of communication. If this is the main/only point of departure from existing literature, then it is not too convincing. Indeed, consider the paper *Byzantine-Robust Distributed Learning: Towards Optimal Statistical Rates*, Yin et al., 2018 (ref [30] in this paper). The authors in this paper do in fact study a *one-shot* robust distributed learning algorithm that requires just one round of communication with the server right at the very end. Thus, the aspect of communication-efficiency/one-shot communication is by no means unique to the eigenspace estimation problem considered in this work. In this context, I would encourage the authors to clarify distinctions relative to prior work.

- It seems that the authors consider a *batch* setting where each node $i$ already has access to multiple samples of $A$, and has been able to construct an estimate $A_i$ that is not too far off from $A$ (at least line 114 alludes to this). Would the ideas/results in this paper extend naturally to a *streaming/online* setting where nodes acquire samples of $A$ over time, and exchange information immediately via the server to compute instantaneous estimates? The challenge that I foresee here is that initially, even for the good nodes, their estimates $A_i$ may be far off from the ground truth $A$ (since they haven't acquired enough samples yet). Hence, it may be hard to separate adversarial errors from statistical errors.

- In this work, the additional price paid due to adversarial corruption is $O(\sqrt{\alpha})$. While such an additive price is in fact unavoidable (based on lower bounds in the robust mean estimation literature), could the dependence on $\alpha$ be improved if the noise is well behaved? To be more precise, suppose the samples of $A$ seen by the good nodes are perturbed by i.i.d. Gaussian noise. In this case, recent results in the high-dimensional robust mean estimation literature reveal that one can go from $O(\sqrt{\alpha})$ to $O({\alpha})$, in terms of the dependence on the corruption fraction $\alpha$; see [R1] below. I am wondering if similar conclusions can be drawn for the eigenspace estimation problem considered here.

[R1] *All-in-one robust estimator of the gaussian mean*, Dalalyan and Minasyan, Annals of Statistics, 2022.

- On a final note, is Algorithm 2 and its analysis in Proposition 1 new? How does the analysis compare with that in Ref [25]?


----------------------------------------------- **Post Rebuttal**  ---------------------------------------------

I thank the authors for the rebuttal. I have read through the authors' responses to both my questions, and the questions raised by the other Reviewers. I think the authors have done a good job highlighting the non-trivial aspects of both the algorithm design and the analysis. In particular, my own opinion is that an approach **should not be penalized** just because it combines/builds on existing algorithmic ideas. There is considerable credit in piecing together existing ideas in a meaningful way, and providing concrete theoretical statements to back up the developed theory - the paper under review falls in this category. As such, I have increased my score.

---

> ### Author Response · Authors · 2022-08-01
> **Response to Reviewer GRdZ**
>
> Thank you for the positive review and the thoughtful feedback!
>
> Please see below for individual responses to your questions and issues raised.
>
> ## Novelty and Byzantine-robust learning literature
>
> We agree on the importance of positioning our work with respect to the Byzantine-robust
> learning literature, which we intend to expand on in the revised manuscript.
> We never claimed that we are unique in pursuing a one-shot distributed learning
> approach, and we apologize for any misunderstanding. We will clarify this in the
> upcoming revision.
>
> What we do believe is unique to our work is the need to resolve ambiguities of
> local solutions before aggregating them. For example, the one-shot distributed
> algorithm in Ref. \[30\] conveniently assumes that all local loss functions are
> strongly convex (in fact, *quadratic*), so that solutions are unique; we make no
> such assumption here.
> Furthermore, while it is possible to formulate distributed PCA as an optimization problem,
> it does not fix the issue of orthogonal ambiguity in local solutions, which appears
> to be unaddressed by existing one-shot algorithms in the Byzantine-robust literature.
>
> ## Batch vs. online setting
>
> This is an excellent question, albeit one that we have had little time to think
> about. In the absence of any additional assumptions, we agree with your intuition
> that there will be an inevitable "burn-in" phase during which the local estimates
> are not accurate enough and thus practically indistinguishable from corrupted
> responses. It might also be interesting (and theoretically feasible) to study the setting where data arrive in batches, but the
> underlying distribution / covariance matrix "drifts" over time, possibly at a controlled rate.
>
> ## Improving the dependence on $\alpha$
>
> This is another excellent suggestion. While it is reasonable to expect that
> well-behaved noise at the sample level could lead to well-behaved noise at the
> eigenvector level, we have limited hope that extending our bounds is easy
> (unless, potentially, the Gaussian noise is added directly to the responses $ V_i $ rather than the matrices $ A_i $). The reason is that,
> even when the noise added to the samples is Gaussian, the resulting eigenspaces
> $ V_i $ are not necessarily equal in distribution to (ground truth) + (Gaussian
> noise).
>
> This observation means that applying existing results in the robust
> stats literature in a black-box fashion might be hopeless. However, we are hopeful that studying the
> statistical properties of the $ V_i $'s (or other related objects) directly, *a la* \[FWWZ19\],
> might lead to interesting insights (for example, it is already known that when
> the samples in distributed PCA are Gaussian, the *spectral projectors*
> $V_i V_i^T$ are unbiased, even though the $V_i$'s themselves are generally biased).
>
> ## Novelty of Alg. 2
>
> The robust reference selection is influenced by the one appearing in Ref. \[25, p. 243-244\] (but not 100% identical). A summary of the main differences follows:
>
> 1. The algorithm from Ref. \[25\] is framed as a method to boost the success probability of a randomized algorithm at a logarithmic cost (the difference with other standard boosting methods in the TCS literature is that the quality of each solution
> cannot be evaluated, but is guaranteed to be "good" with some probability $> 1/2$).
>
> 1. In \[25\], the authors assume that a nontrivial bound on the worst-case accuracy of the "inliers" (or successful trials, in more general language) is known and passed as a parameter to the algorithm. We make no such assumption here.
>
> Finally, note that we do not claim that Algorithm 2 is our invention. However, we
> decided to add a self-contained proof as the language used in Ref. \[25\] is hard
> to parse and relies on terminology introduced in earlier chapters.
>
> ## References
>
> \[FWWZ19\]: Fan et al. *Distributed estimation of principal eigenspaces*, 2019.

---

### Official Review · Reviewer_86Mv · 2022-07-13

**Rating:** 4
**Confidence:** 4
**Soundness:** 3 good
**Presentation:** 3 good
**Contribution:** 2 fair

**Summary:**

Thanks for the rebuttal. It provides some clarity and I have increased my score slightly. However, as can be inferred from the rebuttal, my original assessment of algorithmic and theoretical contribution being quite limited is accurate. The existence of a CVPR paper with an inferior methodology does not warrant acceptance of a paper with limited contribution.

Regarding the spectral norm analysis, as the authors acknowledged, there is a number of papers requiring spectral norm conditions and spectral-norm based analysis. See, e.g.,

a) Steinhardt, J., Charikar, M., and Valiant, G. (2017). Resilience:
A criterion for learning in the presence of arbitrary
outliers

b) Data, D. and Diggavi, S. (2020). Byzantine-resilient SGD
in high dimensions on heterogeneous data.

c) Blanchard, P., Guerraoui, R., Stainer, J., et al. (2017). Machine
learning with adversaries: Byzantine tolerant gradient
descent. In Advances in Neural Information Processing
Systems

d) Bulusu, S., Khanduri, P., Sharma, P., and Varshney, P. K.
(2020). On distributed stochastic gradient descent for
nonconvex functions in the presence of byzantines.

and the references therein and their citations.

-----Original review
This paper proposes an algorithm for the problem of eigenspace estimation algorithm for distributed environments with arbitrary node failures. In particular, a Byzantine adversary model is chosen where a fraction of the participants may communicate arbitrary orthogonal matrices to a server where aggregation happens. The paper builds upon the existing work to put together three components that result in the proposed scheme: i) suitable reference selection, ii) alignment, and iii) robust filtering. It is shown the proposed algorithm enjoys a bounded error consisting of three additive error terms.



**Questions:**

1. What is novel about putting the three components discussed in Algs 2,3,4? What are the algorithmic and theoretical challenges in doing so?
2. To me, the interesting part was the robust reference selection method discussed in Algorithm 2. It is stated that this method builds upon [25]. Could you elaborate on this and discuss the similarities and differences?
3. An iterative refinement method is discussed in [8] to limit the variance stemming from arbitrary reference selection. How does the proposed method without the robust filtering component compare with the iterative refinement method from [8]?

**Limitations:**

yes.

**Strengths And Weaknesses:**

Strength:
1. A specialized algorithm for robust distributed eigenspace estimation and PCA.

Weakness:
1. Limited contribution and significance. Although the paper may be technically sound, the contribution seems to be a mere application of three existing components put together in a seemingly non-difficult fashion.
2. Limited experiments. The impact of changing important parameters such as the number of samples $n$ and number of nodes must be studied. Furthermore, the performance against the centralized solution must be compared and other attach mechanisms must be tested (the latter is important to test the efficacy of the robust reference selection method).

---

> ### Author Response · Authors · 2022-08-01
> **Response to Reviewer 86Mv**
>
> Thank you for your time and thoughtful feedback. We address your
> questions and concerns individually below.
>
> ## Novelty of Algorithm 1
>
> Regarding the novelty of Algorithm 1, we kindly ask you to refer to our response to Reviewer z2xR [here](https://openreview.net/forum?id=g-I_qqceH2n&noteId=xlMh8AKVVqY). We are happy to provide additional clarifications where needed!
>
>
> ## Robust reference selection and Ref. \[25\]
>
> We are excited to hear you were interested in the robust reference selection algorithm! The selection algorithm is influenced by the algorithm sketched in Ref. \[25, p. 243-244\] (but is not 100% identical). A summary of the main differences follows:
>
> 1. The algorithm from Ref. \[25\] is framed as a method to boost the success probability of a randomized algorithm at a logarithmic cost (the difference with other standard boosting methods in the TCS literature is that the quality of each solution
> cannot be evaluated a priori, but is guaranteed to be "good" with some probability $> 1/2$).
>
> 1. In \[25\], the authors assume that a bound on the worst-case accuracy of the "inliers" (or successful trials, in more general language) is known and passed as a parameter to the algorithm. We make no such assumption here.
>
> Finally, note that we do not claim that Algorithm 2 is our invention. However, we
> decided to add a self-contained proof as the language used in Ref. \[25\] is hard
> to parse and relies on terminology introduced in earlier chapters.
>
> ## Iterative refinement method of Ref. \[8\]
>
> This is an excellent suggestion. While the method from Ref. \[8\] is a natural
> candidate for variance reduction, one would expect that the bias introduced by
> adversarial perturbations would dominate.
>
> However, upon putting this intuition to the test numerically, we found that the
> iterative refinement method from Ref. \[8\] essentially performs comparably to
> robust reference selection when the robust mean estimation step is omitted
> entirely. However, both methods are uniformly worse than Algorithm 1 from the
> manuscript, which includes the robust aggregation step. **We have included the additional
> plots in the updated supplementary material**.
>
> While the above findings suggest that adversarial perturbations may not affect
> the reference selection algorithm so adversely, we are not aware of any provable
> upper bounds on the performance of the iterative refinement method in \[8\], which
> was introduced as a heuristic method - in contrast, Algorithm 2 in our manuscript is
> accompanied with guarantees on the quality of the solution. We also do not want to
> rule out the possibility that a different corruption configuration exists, for which
> the iterative refinement method of Ref. \[8\] performs worse that our robust selection
> method.
>
> ## Other attack mechanisms
>
> We agree with the suggestion that attack mechanisms othert than the one described
> in Section 3.1 should be studied. However, we believe the attack mechanism we employ
> in that section is in some sense the hardest one to protect against (and haven't been able to identify a similarly powerful alternative).
> Indeed, the robust statistics literature has identified the setting where multiple outliers are "conspiring"
> by corrupting their responses towards the same "direction" as the most challenging; for a reference, see e.g., the recent survey by Diakonikolas and Kane \[DK19\].
>
> In any case, we are always happy to consider any alternative attack mechanisms you might have in mind and that we possibly missed during our reading of the robust statistics literature.
>
> ## References
>
> \[DK19\]: I. Diakonikolas, D. Kane. *Recent Advances in Algorithmic High-Dimensional Robust Statistics*, 2019.

---

> ### Author Response · Authors · 2022-08-09
> **Follow-up response to Reviewer 86Mv**
>
> Thank you for the response and pointing us to the additional references, as well as the
> bump in score. Excuse the brief response (due to timing issues) below.
>
> ## References on Byzantine-robust SGD
>
> We understand that pointing to the CVPR 2022 paper might have been a bit of an
> overgeneralization, and the additional references cited in your response are relevant
> and will be included in the revised manuscript. However, we would like to point out that
> these additional references also focus on multi-round algorithms (in particular, SGD)
> which are not attractive for the distributed eigenspace estimation problem from a communication point of view.
>
> ## On resilience by Steinhardt et al.
>
> Thank you for bringing the paper by Steinhardt et al. to our attention. Our understanding
> based on a quick read of that paper is the following:
>
> 1. While it is true that this paper focuses on norms other than Euclidean, its results are not immediately applicable to instances where samples are matrix-valued (unless
> one adopts the same flatten-matrix-to-vector approach that we described in our original response). Indeed, it seems that Algorithms 1 and 2 as well as several technical
> conditions used in the paper require the samples $x_i$ to be $d$-dim. vectors. We suspect that immediate extensions (i.e., using the existing results as a black-box) to the matrix-valued setting will be limited to matrix norms that can be
> expressed as row-wise or column-wise operations on vector norms (e.g., the $2\to\infty$ norm of a matrix).
>
> 2. Section 5.2 contains a matrix recovery result for robustly approximating a rank-$k$
> projection of a set of $n$ sample vectors (a $\delta$-fraction of which is arbitrarily corrupted). While this can encode *sample*-level corruptions in robust PCA, it is
> not clear to us how it can be adapted to our setting. An additional (although less important) difference is that the algorithm provided does not return an *exact* rank-$k$ approximation, whereas distributed eigenspace estimation focuses on exact rank-$k$
> responses even in the non-robust setting.

---

### Official Review · Reviewer_z2xR · 2022-07-14

**Rating:** 6
**Confidence:** 4
**Soundness:** 3 good
**Presentation:** 3 good
**Contribution:** 3 good

**Summary:**

This paper studies distributed PCA. Let A be the matrix of interest. Each machine has access to a local version of A_i, and the goal is to design communication-efficient algorithms to estimate the eigenspace of A. This is a classical problem, and the contribution of this paper falls into a new algorithm that is robust to outliers in A_i. It is shown that the approximation error matches a recent non-robust algorithm up to a certain additive error.

**Questions:**

See above. In particular, I feel it is a serious technical flaw to have a failure probability = $1/n + \log(mn)p$, which makes sense only when the problem size stays as a constant.

**Ethics Review Area:**

["I don’t know"]

**Strengths And Weaknesses:**

- The term `local matrix' was used in many places but its definition remains unclear. The only example was given at line 23, and a formal restriction was set out in Eq. (6). It is, however, unclear to me when (6) is satisfied under standard assumptions on the clean data.

- Theorem 1: the presentation here is too general to understand the merit. For example, the scaling of $E_{high}$ is unclear. While it can easily be upper bounded by $(\delta_r / \delta)^2$, the latter itself is unbounded. Readers have to read 4 more pages to see that it is a vanishing term (under certain conditions).

- Theorem 4 and Corollary 1: it is surprising that the failure probability does not vanish - it even grows with problem size! I guess this is the price paid to make the approximation error diminish, but overall such tradeoff is inherently making the results vacuous.

- The Algorithm 1 seems a sandwich of prior results: step 1 is borrowed from [25], step 2 is from [8], and step 3 (robust mean estimation) has been broadly studied in recent years. So what is the main technical contribution?

---

> ### Author Response · Authors · 2022-08-01
> **Response to Reviewer z2xR (Part I)**
>
> We appreciate the thoughtful review and careful read. We address the issues you
> raise individually below:
>
> ## On the failure probability
>
> **It is not true that the error bound is vacuous**. Since $ p $ is a parameter
> that trades off success probability and error bound, and is generally freely
> chosen in $(0, 1/2)$. Thus, one could trivially substitute
>
> $$
> 	p \gets \frac{p'}{\log(mn)}
> $$
>
> to obtain a failure probability of $ \frac{1}{n} + p' $. This substitution
> translates in the following increase in the error:
> $$
> \frac{\log(1 / p)}{m} \quad \text{becomes} \quad \frac{\log(\log(mn) / p')}{m}.
> $$
> This is a negligible increase given that the extra cost is an additive
> $\frac{\log \log(mn)}{m}$.
>
> In fact, the same strategy could lead to a failure probability of $\frac{2}{n}$,
> which clearly goes to $0$ as the sample size $n$ increases, at the cost of an additional $\frac{\log(n)}{m}$ factor.
>
> ## On the term 'local matrix'
>
> The term 'local matrix' refers to the matrices $ A_i \in \mathbb{R}^{d \times d} $
> that are observed by each of the non-compromised machines. These matrices can be
> arbitrary symmetric matrices, as long as they satisfy the error bound in Assumption 2.
>
> The error bound in Eq. (6) is a standard error bound in the literature
> (see, e.g., \[CBD21, FWWZ19\] for works on distributed PCA and \[Stew73\] for a classical
> reference in the literature on matrix perturbation theory)
> and it is satisfied in distributed PCA under subgaussian designs with high probability,
> as per Section 3 in our paper.
> Intuitively, this bound ensures that the local solutions $ V_i $ are
> indeed approximating the **leading** (as opposed to an arbitrary set of) $ r $
> eigenvectors. In a practical setting where, e.g., every local machine uses an
> eigensolver such as $\texttt{eigs}$, this would be a minimal assumption that
> ensures the eigensolvers converge to the "correct" set of eigenpairs, rather
> than an arbitrary set of eigenpairs for which the stopping residual is small.
>
> ## On the scaling of $ E_{\text{high}} $
>
> As we write in the manuscript, $ E_{\text{high}} $ contains *high-order* errors,
> the sum of which will be dominated by $ E_{\text{oracle}} $ in "typical" situations; i.e., when all the inliers are producing good approximations so that
> $ \\| A_i - A \\|_2 / \delta \ll 1 $. We will make sure to emphasize this in the
> revised manuscript.
>
> We understand that the presentation might seem too general given that Theorem 1
> is only applied to distributed PCA in our manuscript. However, we felt it is
> important to separate the different components of the error bound for the
> following reasons:
>
> 1. **Interpretation**: each of the error terms comes from a different source. The
>    $ E_{\text{oracle}} $ part is the "best-possible" as it matches that of a
>    centralized estimator. The term $E_{\text{high}}$ owes to the fact that we
>    are resolving the orthogonal ambiguity of the problem with limited
>    information, and appears in standard references such as
>    \[CBD21, FWWZ19\]. The last term,
>    $ E_{\text{robust}} $, quantifies the effect of corruptions; as Reviewer GRdZ
>    points out, its scaling as a function of the corruption fraction $\alpha$ is
>    consistent with well-known error bounds in the robust statistics literature.
>
> 1. **Applications**: It is important to note that our error bound is application-agnostic and deterministic (the only source of failure probability
> is randomization in the robust mean estimation component).
>    It applies to any robust distributed eigenspace estimation problem, and can
>    potentially be applied to other Byzantine-robust settings where local
>    solutions are subject to orthogonal ambiguities but not necessarily eigenvector matrices. Thus, the merit in having a generic error bound is
>    that it facilitates its use in a black-box fashion in future work where
>    individual terms may be easier to control than a single opaque error bound.
>
> **Please read [below](https://openreview.net/forum?id=g-I_qqceH2n&noteId=xlMh8AKVVqY) for the second part of our response, addressing the novelty of Algorithm 1 in the manuscript.**
>
> ## References
>
> \[CBD21\]: Charisopoulos et al. *Communication-efficient distributed eigenspace estimation*, 2021.
>
> \[FWWZ19\]: Fan et al. *Distributed estimation of principal eigenspaces*, 2019.
>
> \[Stew73\]: G. W. Stewart. *Error and perturbation bounds for subspaces associated with certain eigenvalue problems.*, 1973.

---

> > ### Author Response · Authors · 2022-08-01
> > **Response to Reviewer z2xR (Part II)**
> >
> > ## Novelty of Algorithm 1
> >
> > We spent considerable time and effort making Algorithm 1 as "modular" as
> > possible, so we are in fact delighted to read that it is easy to digest
> > and all the individual components seem transparent.
> > However, we would not consider it a "trivial" algorithm to come up with or
> > analyze. There are multiple technical steps involved, which we will try to
> > summarize below:
> >
> > ### Robust reference selection
> >
> > Algorithm 2 in the manuscript is an adaptation of an existing robust distance
> > estimator from the optimization literature. However, even with this algorithm
> > at hand, it is not obvious that aligning with $V_{\mathsf{ref}}$ is sufficient
> > (note that $V_{\mathsf{ref}}$ could even be an outlier!). One still needs to show
> > that averaging over inliers that have been aligned with $V_{\mathsf{ref}}$ is,
> > in some sense, equivalent to averaging over inliers aligned with the ground truth
> > $V$.
> >
> > To do so, we combine a path-independence result from \[Stew12\] with
> > a "reverse" Davis-Kahan style theorem (provided in the supplement). The key to
> > pushing this through is to view the aligned estimates $V_{i}^{\mathsf{corr}}$
> > (in the notation of Lemma 1 in the manuscript) as the eigenvectors arising from
> > a carefully chosen sequence of perturbations to $A$. We maintain that this
> > component of the analysis is highly nontrivial.
> >
> > ### Robust mean estimation
> >
> > After the local estimates have been aligned with a robust reference solution, the
> > remaining question is how to aggregate them at the central node. There are
> > several choices here: naive averaging, coordinate wise / geometric median, and robust mean estimation.
> >
> > While robust mean estimation appears an easy
> > choice in hindsight, existing analyses of the iterative filtering algorithm in
> > the literature always focus on the Euclidean norm, which is the Frobenius norm
> > in the case of matrix-valued inputs. However, this is equivalent to "flattening"
> > all eigenvector matrices and viewing them as $(dr)$-dimensional vectors, which
> > has the following drawbacks:
> >
> > - The resulting bounds depend on a very unnatural empirical covariance matrix (the empirical covariance matrix of flattened inputs), whose norm is hard to
> > bound in terms of problem data (e.g., in robust PCA).
> > - The error rate we obtain is with respect to the Frobenius norm, while we really do care about the spectral (i.e., $\ell_2 \to \ell_2$ operator) norm.
> >
> > To avoid this, we analyze the iterative filtering algorithm focusing on spectral
> > norm bounds and with a crucial modification in the sample covariance matrix used
> > in each step; in particular, inputs are preserved in their original shapes
> > instead of flattened. This is the key modification that enables
> > a spectral norm error bound free of unnecessary $\sqrt{r}$ muliplicative factors that would arise if norm equivalences were used.
> >
> > Finally, we make two additional remarks with respect to novelty:
> >
> > 1. To our knowledge, our analysis of the robust mean
> > estimation algorithm is a departure from the robust statistics
> > literature, in the sense that it produces error bounds in norms that are not
> > the usual Euclidean norm. In this comparison, we exclude works such as
> > \[LM18\] which work with arbitrary norms but rely on inefficient (i.e., exponential runtime) algorithms. Nevertheless, we are not making this claim
> > with absolute confidence since we are not experts in this area and the existing
> > literature is vast.
> >
> > 1. Even though robust mean estimation seems obvious as an aggregation step in
> > hindsight, it is often overlooked (instead, one often encounters statistically
> > inefficient algorithms such as the coordinatewise median, which
> > degrade by a dimension-dependent factor given worst-case inputs). For example, a
> > recent CVPR 2022 paper \[Kim22\] proposes a diffusion-based aggregation algorithm
> > for robust federated learning that (i) comes with no guarantees and (ii) is not
> > benchmarked against the more natural choice of robust mean estimation.
> >
> > ## References
> >
> > \[CBD21\]: Charisopoulos et al. *Communication-efficient distributed eigenspace estimation*, 2021.
> >
> > \[FWWZ19\]: Fan et al. *Distributed estimation of principal eigenspaces*, 2019.
> >
> > \[Kim22\]: K. I. Kim. *Robust Combination of Distributed Gradients Under Adversarial Perturbations*, 2022.
> >
> > \[LM18\]: G. Lugosi, S. Mendelson. *Near-optimal mean estimators with respect to general norms*, 2018.
> >
> > \[Stew12\]: G. W. Stewart. *Smooth local bases of perturbed eigenspaces*, 2012.

---

> > > ### Comment · Reviewer_z2xR · 2022-08-09
> > > **reviewer response**
> > >
> > > Thank you for the clarification. The author response addressed my main concern on the non-vanishing failure probability. While leaving a quantity $p$ in the theorem seems to make it general, I feel that at minimum, a remark needs to be added for interesting regimes such as $p = 1/ m$.
> > >
> > > The response also clarifies the novelty in using robust mean estimation for the PCA problem. Although I still feel the presentation leads to confusion (and ironing it requires nontrivial rework), I personally appreciate the technical contributions.
> > >
> > > I am raising my rating from '3' to '6'.

---

> > > > ### Author Response · Authors · 2022-08-09
> > > > **Follow-up response**
> > > >
> > > > Thank you very much for your time and the bump in score.
> > > >
> > > > Given our discussion, we will certainly add a note on setting $p$ appropriately (or possibly just restate the theorem with the probability and error bound we describe in our original response to you) in the revised manuscript.
> > > >
> > > > We are also happy to hear that our response provided clarity on the novelty. We will make our best effort to improve the presentation and iron out the main technical contributions in our revision.

---

### Author Response · Authors · 2022-08-01
**Responses & updates to manuscript**

We would like to thank all reviewers for their time and thoughtful feedback.

We have updated the manuscript and supplementary material to reflect some of the
issues raised by the reviewers. The supplement contains a new section with expanded numerical experiments; changes in both files are marked in blue for the reviewers' convenience.

Below, we provide individual responses to each reviewer.

---

### Meta-Review · Area_Chair_ywL7 · 2022-08-20

**Recommendation:** Accept
**Confidence:** Less certain

**Metareview:**

The scores on this paper are mostly positive, with only one being below the threshold.  That reviewer was primarily concerned about novelty and significance, which is a rather subjective matter, and at least one reviewer defended the paper in this regard (and all 3 other reviews indicated it to be sufficient).  Most other concerns appear to have been generally resolved during the rebuttal period.

Even the reviewer that defended the novelty/significance did acknowledge that the presentation could be significantly improved, particularly for making clear what the paper offers compared to the existing literature.  I strongly encourage the authors to carefully consider the presentation of terminology, notation, and discussion of contributions/novelty, so that readers are able to grasp these things as easily as possible.

Overall, while the decision is not quite definite, this paper does appear to pass the bar.

**Award:**

No

---

### Decision · Program_Chairs · 2022-09-14

Accept